# A 4D-Printable Photocurable Resin Derived from Waste Cooking Oil with Enhanced Tensile Strength

**DOI:** 10.3390/molecules29092162

**Published:** 2024-05-06

**Authors:** Yan Liu, Meng-Yu Liu, Xin-Gang Fan, Peng-Yu Wang, Shuo-Ping Chen

**Affiliations:** College of Materials Science and Engineering, Guilin University of Technology, Guilin 541004, China; liuyaner1116@163.com (Y.L.); 2120210291@glut.edu.cn (M.-Y.L.); 15737213667@163.com (X.-G.F.); 2120220368@glut.edu.cn (P.-Y.W.)

**Keywords:** 4D printing, photocurable resin, waste cooking oil, tensile strength, shape memory performance

## Abstract

In pursuit of enhancing the mechanical properties, especially the tensile strength, of 4D-printable consumables derived from waste cooking oil (WCO), we initiated the production of acrylate-modified WCO, which encompasses epoxy waste oil methacrylate (EWOMA) and epoxy waste oil acrylate (EWOA). Subsequently, a series of WCO-based 4D-printable photocurable resins were obtained by introducing a suitable diacrylate molecule as the second monomer, coupled with a composite photoinitiator system comprising Irgacure 819 and p-dimethylaminobenzaldehyde (DMAB). These materials were amenable to molding using an LCD light-curing 3D printer. Our findings underscored the pivotal role of triethylene glycol dimethacrylate (TEGDMA) among the array of diacrylate molecules in enhancing the mechanical properties of WCO-based 4D-printable resins. Notably, the 4D-printable material, composed of EWOA and TEGDMA in an equal mass ratio, exhibited nice mechanical strength comparable to that of mainstream petroleum-based 4D-printable materials, boasting a tensile strength of 9.17 MPa and an elongation at break of 15.39%. These figures significantly outperformed the mechanical characteristics of pure EWOA or TEGDMA resins. Furthermore, the EWOA-TEGDMA resin demonstrated impressive thermally induced shape memory performance, enabling deformation and recovery at room temperature and retaining its shape at −60 °C. This resin also demonstrated favorable biodegradability, with an 8.34% weight loss after 45 days of soil degradation. As a result, this 4D-printable photocurable resin derived from WCO holds immense potential for the creation of a wide spectrum of high-performance intelligent devices, brackets, mold, folding structures, and personalized products.

## 1. Introduction

As one of the progressive trends within the realm of traditional three-dimensional (3D) printing, four-dimensional (4D) printing technology introduces an additional temporal dimension, empowering printed products to attain self-assembly, adaptability, and even self-repair capabilities [1,2,3,4]. The cornerstone of 4D printing lies in the utilization of intelligent biomimetic printing consumables, which empower printed objects to undergo changes in their shapes or properties in response to external environmental stimuli, such as stress [5], temperature [6], moisture [7], pH [8], and electricity [9]. Among the array of smart materials available, photocurable shape memory polymers (PSMPs) are prominently employed as 4D-printable consumables in light-curing technologies [10,11,12]. They boast remarkable attributes, including high printing precision, superb resolution, and an aesthetically pleasing finish for the printed products. Consequently, these PSMPs exhibit impressive profit margins in the market (>100%) and hold substantial promise in diverse applications, spanning biomimetic design, biomedical scaffolding, smart devices, origami structures, self-lubricating constructs, and beyond [13,14].

In the past half-decade, there has been a significant surge in the research and development of various photopolymerizable resins tailored for 4D printing applications. Predominantly, these resins have leaned towards the utilization of petroleum-based acrylates [15,16,17], acrylamides [18,19,20], or ethylene glycols [21,22,23] as their primary prepolymers. Nevertheless, the continuous advancement of photocurable resins for 4D printing faces formidable challenges centered around cost-effectiveness, sustainability, and environmental responsibility. One of the foremost impediments is the exorbitant expense associated with petroleum-based 4D-printable photocurable resins, with raw material costs often exceeding $20 per kilogram. This financial burden is particularly detrimental when envisioning large-scale commercial applications [24,25]. Furthermore, the escalating momentum towards sustainable development and ecological economics has engendered stricter regulations governing the use of petroleum-derived raw materials. This regulatory shift intensifies the urgency to explore alternative, renewable materials sourced from biomass, such as cellulose [26,27,28], lignin [29,30,31], and vegetable oil [25,32]). Among the spectrum of bio-based polymers under consideration, photocurable resins based on soybean oil (SO) exhibit an array of characteristics that seamlessly align with principles of environmentally responsible practices. These soybean oil-based resins tout attributes including cost-effectiveness, reliance on renewable energy sources, and an exceptional level of biocompatibility when compared to their petroleum-based counterparts. As a result, these resins hold immense promise for diverse applications spanning UV photocurable coatings, adhesive formulations, and the dynamic realm of 4D printing [32,33,34,35]. To illustrate this potential, Miao et al. exemplified the application of a soybean oil-based photocurable resin in 4D printing, with epoxidized soybean oil acrylate (ESOA) serving as the key monomer. Remarkably, this resin showcased thermally induced shape memory properties, enabling it to assume a temporarily fixed shape at −18 °C and seamlessly revert to its original configuration at human body temperature (37 °C). This innovative feature opens up exciting avenues for the development of biocompatible scaffolds, marking a significant leap forward in the field of 4D printing [36].

On the other hand, the waste cooking oil (WCO) is a major type of food waste that contains a large amount of hazardous substances, which can be detrimental to the environment and people’s health without proper handling [37,38]. Thus, converting WCO into beneficial products can not only eliminate potential hazards in regard to food safety, health and the environment, but also bring economic benefits, increase regional income, and create job opportunities [39]. Over the past two decades, significant advancements have been made in recycling and harnessing WCO, leading to the creation of diverse products such as biodiesel [40,41], biolubricants [42,43,44], alkyd resins [45], bioplasticizers [44,46,47], cleaning agents [48,49], and other related items [50,51,52]. However, it is worth noting that conventional products derived from WCO often yield narrow profit margins, often falling below 20% and thus necessitate robust policy and financial backing. For instance, the production of biodiesel from WCO, considered the most effective method of utilization, demands a stable and optimized supply chain for raw materials, intricate pre-treatment of the raw oil, expensive catalysts, extensive production facilities, and substantial investments [53,54,55]. Consequently, this approach is primarily suitable for economically developed regions. To address this pressing issue, in our early-stage efforts, we have endeavored to convert waste cooking oil (WCO) into novel products with a relatively smaller market scale but higher added value and technological sophistication. These products serve as valuable complements to the WCO-to-biodiesel conversion pathway, especially suited for small-scale and cost-effective industrialization. These new products encompass 4D-printable photocurable resin [13], photocurable coating [56], wax [57,58], solid alcohol [59], and more. Compared to the traditional WCO-to-biodiesel recycling approach, these products follow a streamlined synthesis process, mitigating the risk of secondary pollution, maintaining low synthesis costs, and yielding a significant product profit margin. Consequently, they effectively address the prevailing issues associated with the low added value and limited market competitiveness of recycled WCO products. This may present promising commercial opportunities, particularly in economically underdeveloped regions, and introduces novel approaches to WCO recycling. In theory, among the various novel products derived from WCO mentioned above, 3D [60] or 4D [13] printing materials based on WCO exhibit the highest profit margins and the most favorable market competitiveness. Additionally, a 4D-printable photocurable resin developed from WCO not only meets the criteria of cost-effectiveness, sustainability, and environmental friendliness but also avoids conflicting with the demands of food production. As a result, it has the potential to become a prominent trend in the future of 4D-printable consumables.

Our initial experiments indicate that converting WCO into functional 4D-printable materials requires addressing specific challenges within the WCO feedstock. Initially, WCO contains significant impurities, including odor and color contaminants. Additionally, the molecular structure of glycerol triesters in WCO, with its flexible configuration and low double-bond density, could significantly affect the final product’s quality and performance. To overcome these challenges, we employed a modification strategy called “epoxidation and esterification and combination” for WCO. Firstly, WCO underwent epoxidation to produce epoxy waste cooking oil (E-WCO), effectively removing impurities and converting double bonds into epoxy groups. E-WCO then underwent ring-opening esterification with methacrylic acid (MAA), resulting in a photocurable monomer called epoxy waste oil methacrylate (EWOMA). Subsequently, other monomers were introduced and combined with EWOMA. This combination step is crucial as it aims to address inherent issues in pure acrylate-based WCO, like EWOMA. Despite its rapid photopolymerization capabilities, EWOMA alone is not suitable for standalone use in 4D-printable materials due to its high viscosity and soft mechanical characteristics. This viscosity issue, stemming from its high molecular weight, makes it difficult for EWOMA to flow smoothly during printing, potentially leading to defects. Furthermore, the cured product of pure acrylate WCO often lacks mechanical strength due to its highly flexible molecular configuration and low density of unsaturated double bonds. To address these shortcomings, we introduced suitable second monomers like 2-phenoxyethyl acrylate (PHEA) and MAA. These not only reduce the resin’s viscosity but also enhance mechanical properties by creating polymeric segments within the curing resin. This arrangement forms a network that enhances the mechanical properties and overall characteristics of the WCO segment.

Utilizing this approach, we synthesized a resin based on WCO that incorporated EWOMA derived from WCO, as well as PHEA and MAA as supplementary monomers [13]. This WCO-based resin exhibited notable characteristics, including exceptional flexibility (elongation at break: 230.1%), functional pressure-sensitive adhesion, and proficient thermally induced shape memory capabilities. These properties position it as a potential multifunctional polymer consumable suitable for 4D printing applications. However, it is worth noting that the tensile strength of this WCO-based resin (0.48 MPa) falls significantly short when compared to other petroleum-based 4D-printable photocurable resins reported in the literature (mostly in the range of 2–60 MPa) [61,62,63,64,65]. For instance, Chung et al. employed N-Vinylpyrrolidone (NVP), O-polyethylene glycol diacrylate (PEGDA), and glycidyl methacrylate-grafted polyetherimide (PEI-GMA) in the fabrication of 4D-printable photopolymers, achieving a tensile strength of 4.3 MPa [62]. In a separate study, Yang et al. utilized cyclic trimethylolpropane formal acrylate (CTFA) and 2-phenoxyethyl acrylate (PHEA) for the synthesis of a 4D-printable photosensitive resin, resulting in a tensile strength of 2.8 MPa [65]. As a result, its use is limited in applications that require the ability to withstand substantial forces. Consequently, to meet the prevailing commercial demand, the development of a WCO-based 4D-printable photocurable resin with increased stiffness and toughness is imperative.

Based on the results of our previous work, it was observed that the introduction of a second monomer has a significant impact on the performance of WCO-based resins. In comparison to the PHEA molecule utilized in our previous research, diacrylate molecules like triethylene glycol dimethacrylate (TEGDMA) can generate a polymeric network with a much higher crosslinking density, potentially resulting in improved mechanical properties, especially the tensile strength. Therefore, in this paper, we designed and synthesized a series of 4D-printable photocurable resins composed of acrylate-based WCO such as EWOMA or epoxy waste oil acrylate (EWOA), along with different diacrylate molecules (See Figure 1). By employing suitable combinations of monomers, this WCO-based resin demonstrated favorable mechanical strength comparable to mainstream petroleum-based 4D-printable materials as well as good thermally induced shape memory performance and biodegradability. Additionally, the effects of acrylate WCO molecules and diacrylate molecules on the properties of the WCO-based resin are discussed in detail.

## 2. Result and Discussion

### 2.1. Structural Characterization

The infrared (IR) spectra of the liquid WCO-based resin and its 4D-printed product are illustrated in Figure 2a. In the case of the EWOA-TEGDMA resin (A2 sample), the uncured liquid resin displayed prominent absorption peaks attributed to the unsaturated double bonds within EWOA and TEGDMA. Specifically, the stretching vibration of the C=C double bond was observed at 1635 cm^−1^ and 1598 cm^−1^, while the in-plane and out-of-plane bending vibrations of =CH_2_ were detected at 1405 cm^−1^ and 1046 cm^−1^, respectively. Furthermore, absorption peaks at 1296 cm^−1^ and 813 cm^−1^ corresponded to the in-plane and out-of-plane bending vibrations of –CH=, respectively. In the 4D-printed product, all the absorption peaks associated with the aforementioned unsaturated double bonds either vanished or significantly diminished. This unequivocally indicated that the light-curing process of this particular WCO-based resin primarily resulted from the polymerization of the C=C double bonds found in EWOA and TEGDMA.

On the contrary, within the cured resin’s polymeric network, various functional groups, including hydroxyl and carboxylate groups, were identified. A broad peak centered at 3439 cm^−1^ signified the stretching vibrations of O–H in the hydroxyl and carboxyl groups, while the peaks at 1731 cm^−1^ corresponded to the vibrations of C=O in the carboxylate group. The presence of these hydroxyl and carboxylate groups was further corroborated by the X-ray photoelectron spectroscopy (XPS) spectra of the 4D-printed product (see Figure 2b–d). In the high-resolution C1s spectrum, characteristic peaks at 284.73 eV, 285.75 eV, 286.43 eV, and 288.86 eV were attributed to the C–C bond, C–O bond in the hydroxyl group, C=O bond in the carboxylate group, and –COOH in the carboxyl group, respectively (see Figure 2c). Similarly, the high-resolution O1s spectra revealed the presence of carboxylate, carboxyl, and hydroxyl groups, marked by characteristic signals at 532.14 eV, 532.72 eV, and 533.28 eV, respectively (see Figure 2d). Considering the insights from the IR spectra, it can be concluded that the 4D-printed product of the EWOA-TEGDMA resin formed a crosslinked polymeric network composed of segments from WCO and TEGDMA. Similar observations were made for the EWOMA-TEGDMA resin (see Appendix A in ESI).

### 2.2. Mechanical Properties

The 4D-printable photocurable resin based on WCO demonstrated superior mechanical strength by incorporating a moderate amount and specific type of diacrylate. As depicted in Figure 3a and Appendix A, the cured product of pure EWOA exhibited very weak mechanical properties, with a tensile strength of 0.03 MPa, making it prone to damage during the 3D printing process. However, by introducing a diacrylate as the second monomer, the mechanical properties of the WCO-based 4D-printable resin could be significantly improved, particularly the tensile strength. For instance, the WCO-based 4D-printable resin containing EWOA and TEGDMA (A2 sample, with a mass ratio of 1:1) demonstrated a commendable tensile strength of 9.17 MPa, which was more than 300 times higher than that of pure EWOA. Furthermore, it was observed that the chain length of the diacrylate played a crucial role in enhancing the mechanical strength. Among the six different diacrylate molecules with varying chain lengths (PEGDA, TPGDA, TEGDMA, HDMA, GDMA, and EGDMA), TEGDMA, with a moderate chain length, exhibited the most significant strengthening effect on the mechanical strength of the WCO-based polymeric network. The tensile strength of the WCO-based resin containing TEGDMA (A2 sample, 9.17 MPa) was 461%, 8%, 26%, 54%, and 160% higher than those containing PEGDA, TPGDA, HDMA, GDMA, and EGDMA, respectively. A similar trend was observed for impact strength. The EWOA-TEGDMA resin (A2 sample) displayed the highest impact toughness (5.34 J/m^2^), which was 162%, 15%, 79%, 216%, and 354% higher than the resins containing PEGDA, TPGDA, HDMA, GDMA, and EGDMA, respectively (refer to Figure 3b).

As illustrated in Figure 3c, due to its branched structure with significant steric hindrance, the pure EWOA resin’s cured network could not form effective close cross-linking, thus the cured product of pure EWOA exhibited weak mechanical characteristics and was unsuitable for use as a practical 4D-printable consumable. However, by introducing a suitable diacrylate like TEGDMA, a cross-linked EWOMA-TEGDMA polymer network, comprising segments of EWOA and TEGDMA, was formed. The polymeric TEGDMA segment exhibited stronger binding force than the EWOA segment, effectively absorbing energy and enhancing the polymer network’s resistance to external stress. Furthermore, TEGDMA with a moderate chain length led to an optimized cross-linking density (2.4 × 10^−3^ mol/cm^3^, see Appendix A), which further stabilized the polymer network, substantially increasing the tensile strength and resulting in a WCO-based resin with moderate rigidity and toughness. In contrast, other diacrylate molecules did not display such a significant reinforcing effect. For example, a diacrylate with an excessively long chain length (like PEGDA) might not effectively absorb external energy, resulting in a relatively lower cross-linking density (1.7 × 10^−3^ mol/cm^3^). Consequently, resins containing PEGDA exhibited lower tensile strength (1.63 MPa) compared to the EWOA-TEGDMA resin (9.17 MPa). On the other hand, a diacrylate with an excessively short chain length (like EGDMA) could result in a higher cross-linking density (2.10 × 10^−2^ mol/cm^3^), but its short chain length could increase internal stress, leading to excessive deformation or breakage along with the EWOA and increased brittleness of the cured resin. As a result, resins containing EGDMA showed weaker elongation at break (2.11%) and impact toughness (1.18 J/m^2^) compared to the EWOA-TEGDMA resin (elongation at break: 15.39%, impact toughness: 5.34 J/m^2^).

The mechanical strength of the 4D-printable photocurable resin could be controlled by varying the amount of TEGDMA. As shown in Figure 4a and Appendix A, the mechanical properties of the resulting resin initially increased and then decreased with increasing TEGDMA dosage. The WCO-based 4D-printable consumable with the optimal recipe (A2 sample, where EWOA: TEGDMA = 1:1 in mass ratio) exhibited a hard and tough resin, presenting a good tensile strength of 9.17 MPa and a nice elongation at break of 15.39%, even surpassing that of pure TEGDMA resin (tensile strength: 8.40 MPa, elongation at break: 4.39%). Similarly, TEGDMA also showed a strengthening effect on the mechanical properties of the WCO-based resin containing EWOMA monomer (see Figure 4a). The pure EWOMA curing product was a soft and brittle material, with a tensile strength of 0.19 MPa and an elongation at break of 5.76%. With an increase in TEGDMA dosage, the mechanical properties of the resulting resin initially increased and then decreased. The WCO-based 4D-printable consumable with the optimal recipe (MA2 sample, where EWOMA: TEGDMA = 1:1 in mass ratio) demonstrated a good tensile strength of 10.74 MPa and a nice elongation at break of 10.52%, approximately 56.5 times and 1.8 times higher than that of pure EWOMA, respectively. The EWOMA-TEGDMA resin exhibited higher tensile strength but lower elongation at break compared to the EWOA-TEGDMA resin, possibly due to the increased rigidity of methyl groups. In comparison to the highly flexible WCO-based resin composed of EWOMA, PHEA, and MAA previously reported [13], the 4D-printable photocurable resin composed of acrylate WCO and TEGDMA described here displayed lower flexibility but significantly higher stiffness. This further confirms that selecting the appropriate second monomer can effectively alter and regulate the performance of the WCO-based resin, thereby obtaining functional materials suitable for various purposes. Furthermore, compared to other petroleum-based photocurable resins, the resulting WCO-based photocurable resin exhibited mechanical properties comparable to them. Its tensile strength surpassed that of 4D-printable materials constructed from the monomers cyclic trimethylolpropane formal acrylate (CTFA) and PHEA (tensile strength: 2.8 MPa, [64]), as well as materials containing the monomers N-Vinylpyrrolidone (NVP), O-polyethylene glycol diacrylate (PEGDA), and glycidyl methacrylate-grafted polyetherimide (PEI-GMA) (tensile strength: 4.3 MPa, [61]). This indicates that a 4D-printable photocurable resin composed of acrylate WCO and TEGDMA could serve as a substitute for petroleum-based 4D-printable photocurable consumables. Such a resin had the capability to manufacture various products capable of withstanding certain forces (see Figure 4b). Additionally, the introduction of TEGDMA also improved the impact resistance of the resin. For instance, the impact toughness of A2 and MA2 resins reached 5.34 kJ/m^2^ and 5.36 kJ/m^2^, respectively, nearly four times that of pure TEGDMA (1.35 kJ/m^2^) (see Figure 4c).

On the other hand, temperature exerted a significant impact on the mechanical properties of WCO-based 4D-printed products. Specifically, as the temperature decreased from above 0 °C, the mechanical properties of A2 resin showed a slight improvement. At 0 °C, the A2 resin maintained its rigid and tough properties, with a tensile strength of 9.94 MPa and an elongation at break of 23.2%. However, as the temperature further dropped to −40 °C, the A2 resin underwent a complete transformation into a hard and brittle material, exhibiting a tensile strength of 4.64 MPa and a low elongation at break of 2.74% (see Figure 4d and Appendix A). This change in mechanical properties due to temperature variation emphasized the potential value of A2 resin as a shape memory material. Similarly, when compared to room temperature (25 °C), the tensile strength and elongation at break of MA2 resin increased at 0 °C, reaching 15.67 MPa and 18.95%, respectively, showcasing its rigid and tough properties. However, as the temperature continued to decrease to −40 °C, the material gradually became hard and brittle, resulting in a reduction in the tensile strength to 7.99 MPa and the elongation at break to 12.16%. Moreover, excessively high temperatures could impair the mechanical properties of this WCO-based 4D-printed product. For instance, when the testing temperature was elevated to 50 °C, the tensile strength and elongation at break of A2 resin decreased to 3.23 MPa and 11.09%, respectively (refer to Figure 4e and Appendix A).

### 2.3. Shape Memory Property

All the 4D-printed products fabricated from the WCO-based resin exhibit characteristic thermal-induced shape memory properties. As depicted in Figure 5, the A2 resin-based 4D-printed product demonstrates excellent toughness, allowing it to easily deform and assume various shapes at room temperature (e.g., 25 °C, the deformation temperature). The deformed product can be effectively fixed by quenching it at a relatively low temperature (e.g., −60 °C, using a dry ice-ethanol solution). Subsequently, rapid shape recovery occurs when the fixed deformed product was returned to 25 °C. For simple rectangular thin film, the recovery process could be essentially completed within 1 min. However, for more complex structures, the recovery process might be extended.

In the shape memory cycle test (refer to Figure 6a), the WCO-based 4D-printable resin (A2 sample) exhibits favorable shape fixation at −60 °C with a high fixation rate (*R_f_*) of 99.73%, accompanied by good shape recovery at 25 °C with a high recovery rate (*R_r_*) of 98.87% and a maximum shape recovery speed (*V_r_*) of approximately 2.86%/min. Moreover, it demonstrates repeatable shape memory behavior, maintaining relatively high *R_f_* (92.78%) and *R_r_* (89.82%) even after 10 shape memory cycles (see Figure 6c,d). In comparison, EWOMA-TEGDMA resins show relatively weaker shape memory performance compared to EWOA-TEGDMA resins. The MA2 resin exhibits an *R_f_* of 94.29% at −60 °C, an *R_r_* of 95.85% at 25 °C, and a *V_r_* of 2.53%/min (see Figure 6b). However, its *R_f_* and *R_r_* decrease to 82.77% and 76.51%, respectively, after 10 shape memory cycles (see Figure 6c,d). Consequently, EWOA-TEGDMA resin (such as A2 resin) proves to be more suitable as a practical shape memory material than EWOMA-TEGDMA resin.

Additionally, it was observed that the shape memory temperature of the WCO-based resin could be determined based on the results obtained from DSC (see Figure 6e,f). As shown in Figure 6e, the DSC curve of A2 resin exhibited an exothermic peak at −31.9 °C, indicating that below this temperature, the entire polymeric network of the resin was in a “fully frozen” glass state. Therefore, to ensure effective shape fixation, the material should be kept deeply subcooled, namely at a suitable temperature below −31.9 °C. For instance, A2 resin displayed a high *R_f_* (98.33%) when fixed in a dry ice–ethanol solution at −60 °C. Conversely, if a higher temperature (>−31.9 °C) was chosen, some segments of the A2 resin chains thawed partially, leading to incomplete fixation. For example, its *R_f_* decreased to 90.55% and 79.00% at −25 °C and 0 °C, respectively (refer to Figure 6g). On the other hand, during the heating curve, an endothermic peak at 21.65 °C was observed, which could be attributed to the glass transition of TEGDMA and EWOA segments, indicating completely unfrozen segments in A2 resin above this temperature. Therefore, a recovery temperature of 25 °C was preferred to ensure complete shape recovery. For instance, as shown in Figure 6h, A2 resin exhibited a rapid recovery rate during U-shaped testing at 25 °C, with an *R_r_* of 87.5% achieved within 20 s and 99.88% within 180 s. However, when shape recovery was attempted at lower temperatures, some “frozen” WCO and TEGDMA segments were unable to fully move, resulting in lower recovery rates. At −25 °C and 0 °C, the material only achieved *R_r_* values of 30.56% and 58.33%, respectively. Conversely, at higher temperatures (e.g., 50 °C) in hot water, A2 resin could achieve complete recovery at a much faster rate, reaching an *R_r_* of 99.00% within 4 s. However, at this temperature, the mechanical performance of A2 resin significantly decreased (refer to Figure 4d), thus caution was required during operation to avoid product damage.

The shape memory properties of the WCO-based 4D-printable resin were also closely related to its polymer network. As shown in Figure 7, the room temperature (25 °C) was higher than the glass transition temperature of TEGDMA and EWOA segments, so the entire WCO-based polymer network exhibited good elasticity and flexibility. Its original shape could be easily changed under external forces, thereby forming temporary deformation shapes. When products with temporary deformation shapes were cooled at a lower fixed temperature (such as −60 °C), the TEGDMA and EWOA segments could be well “frozen”, and the entire system transformed into a hard and brittle state. From a macro perspective, the temporary deformed shape could not deform again at the fixed temperature, thus achieving good shape fixation. Then, if the product with a fixed deformation shape was placed at a relatively high recovery temperature (such as 25 °C), the TEGDMA and EWOA segments unfroze and regained their elasticity and flexibility, resulting in full shape recovery and completing a shape memory cycle. In comparison to EWOA, EWOMA had an additional methyl group which increased the rigidity of the polymer network. Thus, the cohesive internal stress within the polymer network was larger, making it difficult to achieve fully shape fixation or recovery and making it challenging to ensure high *R_f_* and *R_r_* after multiple shape memory cycles.

### 2.4. Biodegradability

As a 4D-printable material derived from WCO, it was anticipated that the presence of biodegradable glycerol triesters in its polymer network might confer favorable biodegradability characteristics. The biodegradability of the EWOA-TEGDMA 4D-printable material (sample A2) was evaluated through a 45-day soil burial experiment, with commercial 3D printing photocurable resin used as control. As depicted in Figure 8, the commercially available 3D printing resin derived from petroleum feedstocks exhibited resistance to microbial degradation in natural environments. In soil, it experienced only a 0.98% weight loss in the initial 5 days, reaching a degradation rate of merely 1.95% at 45 days, with the rate increase nearing its limit. In contrast, EWOA-TEGDMA resin achieved a degradation rate of 1.98% within the first 5 days and reached 8.34% after 45 days, over four times that of the commercial 3D printing material adhesive, with a continuous acceleration in degradation rate. Combining mechanical strength and cost comparisons of the two resin types (see Table 1), it is evident that, in comparison to commercial 3D printing resin, EWOA-TEGDMA 4D-printable material not only exhibits slightly inferior tensile properties but also offers significantly reduced costs and improved biodegradability, rendering it more sustainable. Additionally, it possesses shape memory capabilities, enabling adaptation to a wider range of applications. Therefore, EWOA-TEGDMA resin holds potential as a substitute for current commercial petroleum-based 3D printing materials. Furthermore, given that approximately 50% of its composition is derived from WCO, it holds significant value in alleviating petroleum resource scarcity, mitigating environmental pollution, and promoting ecological civilization construction.

## 3. Materials and Methods

### 3.1. Materials

The waste cooking oil (WCO) with an iodine value of 83.0 was collected from the three-phase separator of Guilin Bioland Renewable Energy Co., Ltd. (Guilin, China). Based on the result of GC-MS (see Appendix A), the oleic acid showed the highest amount in the triglycerides of the WCO with a mass fraction of 68.3 wt%, which indicated that the raw oil contained a relatively content of unsaturated double bond and was advantageous for enhancing mechanical properties.

Hydrogen peroxide (H_2_O_2_, analytical grade, 30 wt% in H_2_O) and sulfuric acid (98%) were purchased from Xiya Chemical Technology Co., Ltd. (Linyi, China). Glacial acetic acid (99.5%), urea (99%), sodium bicarbonate (99.5%), acrylic acid (AA, 99%, stabilized with 200 ppm 4-methoxyphenol), methacrylic acid (MAA, 98%, stabilized with 250 ppm 4-methoxyphenol), triphenylphosphine (PPh_3_, 99%), hydroquinone (HQ, 99%), triethylene glycol dimethacrylate (TEGDMA, 95%, stabilized with 100 ppm 4-methoxyphenol), poly(ethylene glycol) distearate (PEGDA, average molecular weight ~200, stabilized with 120 ppm 4-methoxyphenol), tripropylene glycol diacrylate (TPGDA 90%, stabilized 120 ppm 4-methoxyphenol), 1,6-hexanediol dimethacrylate (HDMA, 95%, stabilized with 100 ppm 4-methoxyphenol), glycerol dimethacrylate (GDMA, 90%, stabilized 120 ppm 4-methoxyphenol), ethylene glycol dimethacrylate (EGDMA, 98%, stabilized with 110 ppm 4-methoxyphenol), phenylbis (2,4,6-trimethylbenzoyl) phosphine oxide (Irgacure 819, 98%), and p-dimethylaminobenzaldehyde (DMAB, 99%) were all obtained from McLean Company (Shanghai, China). As a control sample, the commercial 3D printing photocurable resin was purchased from Anycubic Technology Co., Ltd. (Shenzhen, China).

### 3.2. Synthesis and 4D Printing of Photocurable Resin Based on WCO

As shown in Figure 1, the synthesis of acrylate-based WCO, encompassing EWOA or EWOMA, employed the “epoxidation and ring-opening esterification” method, as outlined previously [13]. Detailed synthesis steps are provided in Appendix A. The resulting EWOA (or EWOMA) manifests as a brownish-yellow transparent liquid. Both EWOA and EWOMA exhibit photocurability under UV light irradiation with the introduction of an initiator. However, the elevated viscosity of pure acrylate WCO (5271 mPa·s for EWOA and 4322 mPa·s for EWOMA) may present challenges during the 3D printing process, potentially leading to printing defects attributed to flow complications.

To attain practical 4D-printable materials, the acquired EWOA (or EWOMA) monomer underwent a blending process with TEGDMA or alternative diacrylates at a temperature of 60 °C (as detailed in Table 2 and Table 3). Subsequently, Irgacure 819 (as the initiator) and p-dimethylaminobenzaldehyde (DMAB) (as the accelerator) were incorporated at a concentration of 3 wt% relative to the mixed monomers and were diligently stirred at 60 °C until a clear, vibrant yellow solution emerged. Following a cooling period to reach room temperature, a suite of photocurable resins founded on WCO were realized as brilliantly clear yellow liquids. These resins could be swiftly and directly solidified under 405 nm light, rendering them exceptionally suitable for application as consumables in 4D printing. In comparison to the pure EWOA (or EWOMA), the resin containing TEGDMA (such as the A2 and MA2 samples) demonstrated a substantial reduction in viscosity (544 mPa·s for A2 and 159 mPa·s for MA2), facilitating enhanced fluidity throughout the 3D printing procedure and effectively averting printing imperfections (See Appendix A).

The 4D printing of the WCO-based photocurable resin was conducted using a Photon X LCD 3D printer (Anycubic, Shenzhen, China) as previously described in our work [13], which could result in a yellow, transparent product with a smooth surface and clearly defined details (See Figure 1). Detailed printing steps are provided in Appendix A.

### 3.3. Characterization

The evaluation of the photocurable resins derived from WCO was also conducted following the methods outlined in our prior publications [13]. For a comprehensive understanding of the experimental procedures and additional data, kindly refer to the Appendix A.

## 4. Conclusions

In this study, we have presented the synthesis and properties of a WCO-based photocurable resin for 4D printing, composed of acrylate WCO and diacrylate molecules. The experimental results have demonstrated that the WCO-based 4D-printable photocurable resin exhibits remarkable tensile strength when a moderate amount of a specific type of diacrylate is introduced. Among the six diacrylate molecules tested, TEGDMA with a medium chain length has shown the most significant enhancement effect on the tensile strength of the WCO-based polymer networks. The combination of TEGDMA and EWOA in the photocurable resin material results in a rigid and tough material, achieving tensile strength comparable to mainstream 4D-printable materials. Furthermore, it exhibits excellent heat-induced shape memory properties as well as nice biodegradability. Building upon our previous research, the choice of the second monomer can be strategically adjusted to impart desired properties, making it a pivotal element in the development of practical WCO-based photocurable resins. Moreover, the incorporation of an appropriate second monomer offers a heightened degree of versatility to system design. This versatility enables the tailored synthesis of diverse types of WCO-based functional resins to meet various demands. 

In comparison to conventional petroleum-based 4D-printable consumables, this WCO-based alternative emerges as an eco-friendly, cost-effective, and sustainable option. It seamlessly aligns with the evolving trends in 4D printing, offering versatility for producing intelligent devices, brackets, folding structures, and personalized products that require exceptional mechanical performance. Additionally, as a photocurable resin, this material may find potential applications in photocurable coatings or adhesives. Moreover, the conversion of WCO into 4D-printable materials enhances their technical content and added value, as demonstrated in our previous studies. The synthesis process remains straightforward, conducted under mild conditions, providing a convenient route with minimal environmental impact and low production costs. These merits effectively address the challenges associated with the limited added value and market competitiveness of current WCO resource utilization. However, the preparation of WCO-based resin for 4D printing is still at the laboratory research and development stage. In our subsequent research endeavors, we will delve into industrial-scale production processes for such 4D-printable materials. This exploration will serve as the foundation for the commercialization of WCO-based 4D-printable consumables.

## Figures and Tables

**Figure 1 molecules-29-02162-f001:**
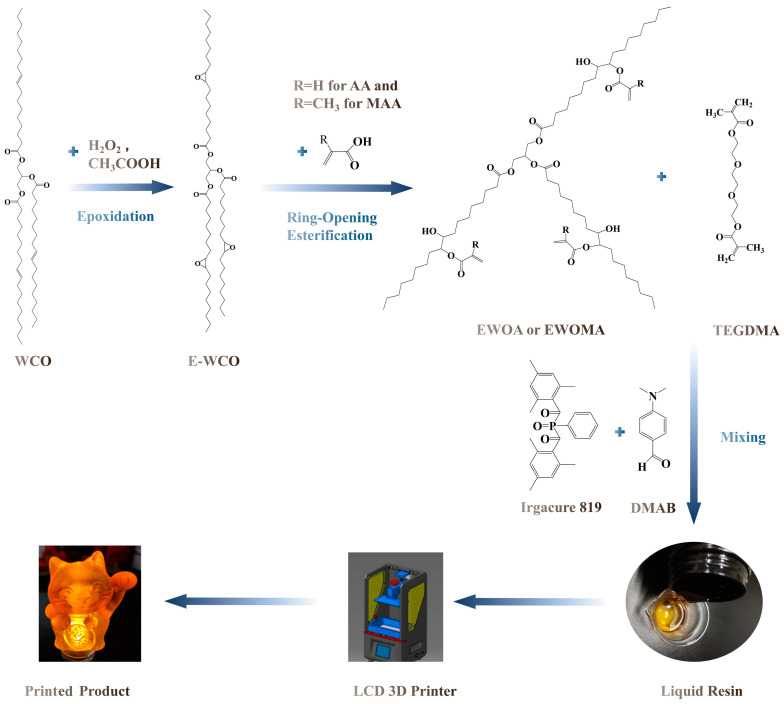
Schematic illustration of the preparation and 4D printing process of the WCO-based photocurable resin.

**Figure 2 molecules-29-02162-f002:**
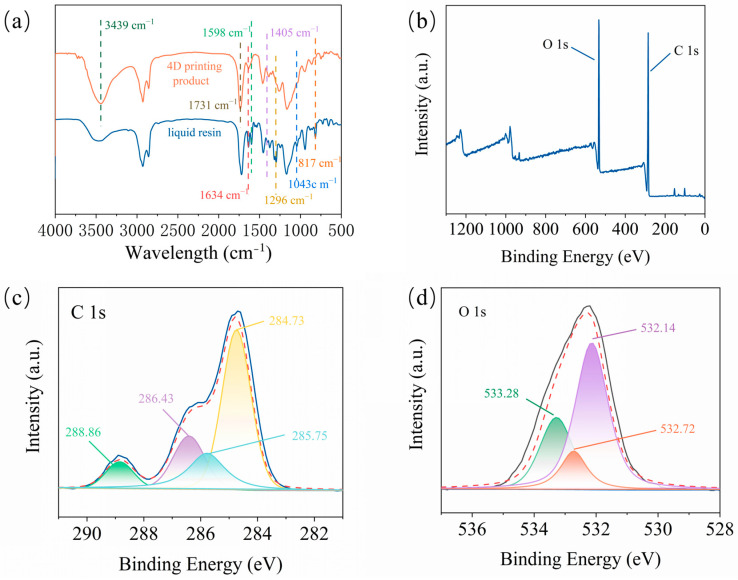
(**a**) IR spectra of 4D-printed product and liquid resin of A2 sample; (**b**–**d**) The full XPS (**b**), C1s (**c**), and O1s (**d**) high-resolution XPS spectra of 4D-printed product of A2 resin.

**Figure 3 molecules-29-02162-f003:**
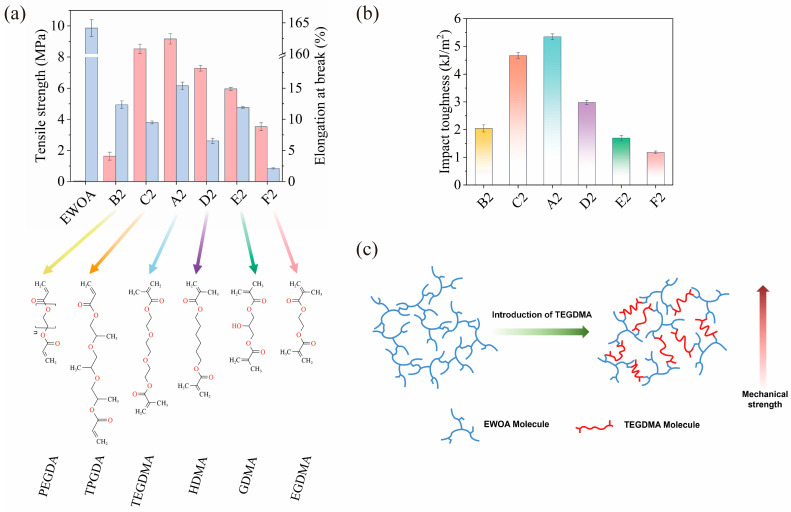
(**a**,**b**) Tensile properties (**a**) and impact toughness (**b**) of pure EWOA and WCO-based 4D-printable resins composed of EWOA and different diacrylate molecules; (**c**) Schematic illustration of the structural differences between the curing products of EWOA-TEGDMA resin and pure EWOA.

**Figure 4 molecules-29-02162-f004:**
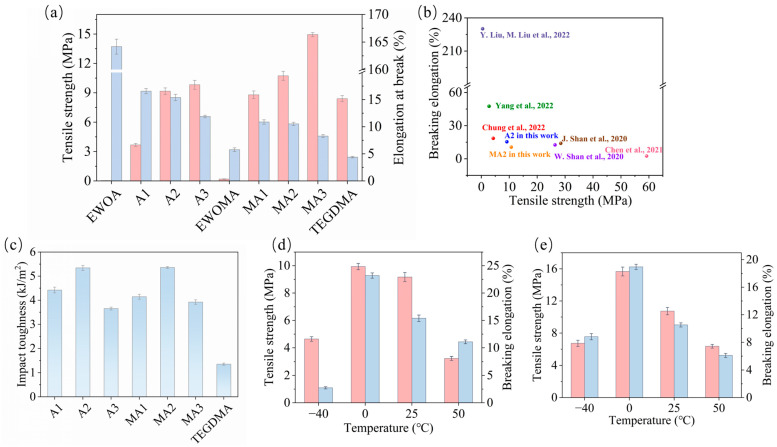
(**a**,**c**) The tensile properties (**a**) and impact toughness (**c**) of EWOA/EWOMA-TEGDMA resins with different dosages of TEGDMA at room temperature; (**b**) The comparison of mechanical properties between the resulting WCO-based resin (A2 and MA2 sample) and other petroleum-based photocurable resins for 4D printing; (**d**,**e**) The tensile properties of the A2 (**d**) and MA2 (**e**) resin at different temperatures.

**Figure 5 molecules-29-02162-f005:**
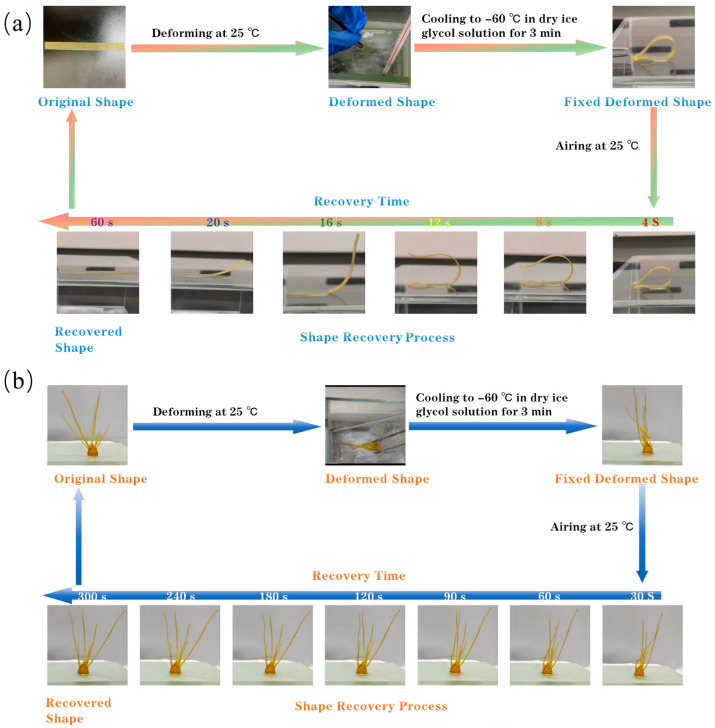
Shape memory cycle of a 4D-printed rectangular thin film (**a**) and radish roots (**b**) of A2 resin.

**Figure 6 molecules-29-02162-f006:**
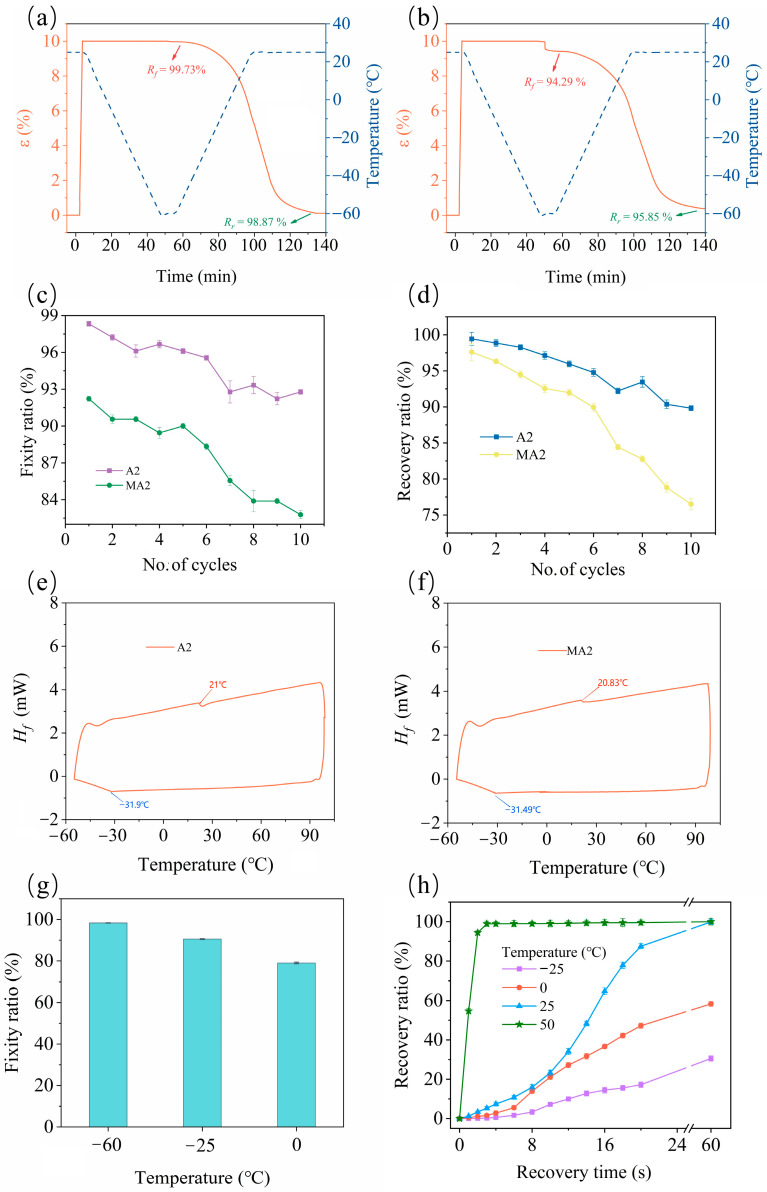
(**a**,**b**) The shape memory behaviour curves of A2 (**a**) and MA2 (**b**) resin; (**c**,**d**) The fixity ratio (*R_f_*, (**c**)) and recovery ratio (*R_r_*, (**d**)) of A2 and MA2 resin within ten shape memory cycles; (**a**–**d**) were all carried out with a fixing temperature of –60 °C and a recovering temperature of 25 °C. (**e**,**f**) The DSC spectra of A2 (**e**) and MA2 (**f**) resin; (**g**) The *R_f_* of A2 resin in water at different fixing temperatures; (**h**) The *R_r_* of A2 resin in water at different recovering temperatures.

**Figure 7 molecules-29-02162-f007:**
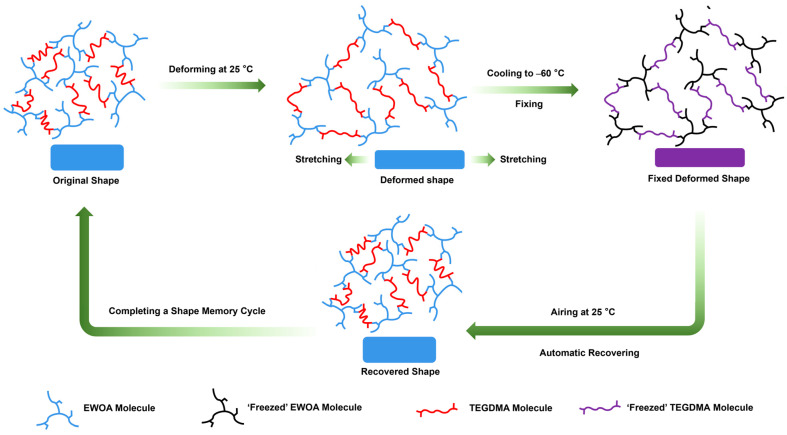
Schematic illustration of the structural transformation in EWOA-TEGDMA resin during a shape memory cycle.

**Figure 8 molecules-29-02162-f008:**
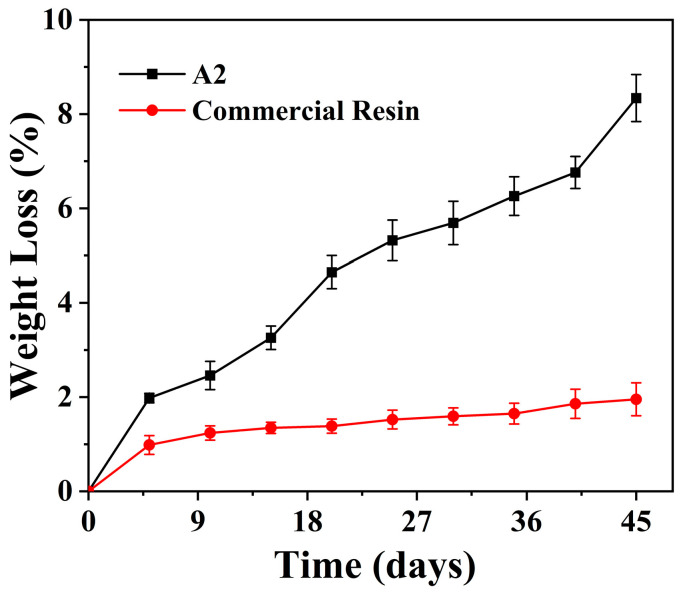
Biodegradation rates of EWOA-TEGDMA resin (A2 sample) and commercial 3D printing resin measured by the weight loss.

**Table 1 molecules-29-02162-t001:** Comparison of EWOA-TEGDMA resin (A2 sample) and commercial 3D printing photocurable resin.

Sample	Tensile Strength (MPa)	Elongation at Break (%)	Biodegradation Rates (45 d Weight Loss, %)	Production Cost ($/kg)
EWOA-TEGDMA resin (A2 sample)	9.17	15.39	8.34	3.0 ^a^
Commercial 3D printing photocurable resin	22.35	15.80	1.95	20.7

^a^ The production cost is estimated based on the quotations of raw material commodities.

**Table 2 molecules-29-02162-t002:** Synthetic formulation of WCO-based 4D-printable resins composed of EWOA and different diacrylate molecules.

Sample	A2	B2	C2	D2	E2	F2
EWOA (g)	100	100	100	100	100	100
PEGDA (g)	-	100	-	-	-	-
TPGDA (g)	-	-	100	-	-	-
TEGDMA (g)	100	-	-	-	-	-
HDMA (g)	-	-	-	100	-	-
GDMA (g)	-	-	-	-	100	-
EGDMA (g)	-	-	-	-	-	100
Irgacure 819 (g)	6	6	6	6	6	6
DMAB (g)	6	6	6	6	6	6

**Table 3 molecules-29-02162-t003:** The synthetic formula of WCO-based 4D-printable resin composed of acrylate WCO (EWOA or EWOMA) and TEGDMA.

Numbering	EWOA (g)	EWOMA (g)	TEGDMA (g)	Irgacure 819 (g)	DMAB (g)
A1	100	0	50	4.5	4.5
A2	100	0	100	6	6
A3	50	0	100	4.5	4.5
MA1	100	0	50	4.5	4.5
MA2	100	0	100	6	5
MA3	50	0	100	4.5	4.5
EWOA (control sample)	100	0	0	3	3
EWOMA (control sample)	0	100	0	3	3
TEGDMA (control sample)	0	0	100	3	3

## Data Availability

Data are contained within the article and Appendix A.

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
