# Peer review of "A 4D-Printable Photocurable Resin Derived from Waste Cooking Oil with Enhanced Tensile Strength"

_molecules, 2024, doi:10.3390/molecules29092162_

Round 1
Reviewer 1 Report
Comments and Suggestions for Authors
Reviewer 2 Report
Comments and Suggestions for Authors
The manuscript presents an approach to recycling waste cooking oil (WCO) by converting it into a photocurable resin for 4D printing applications, with a particular focus on enhancing its tensile strength. The study is motivated by the need for sustainable and cost-effective materials in 4D printing technologies. To strengthen the manuscript further, the authors should consider addressing the following areas for further improvement:
1. While the manuscript highlights the potential of the WCO-based resin, it lacks a detailed comparison with existing 3D printing resin in terms of cost, sustainability, and performance metrics.
2. The manuscript could benefit from additional structures like rectangular bar shape sample, images showcasing the final printed objects and demonstrating their 4D capabilities and recovery.
3. In the Results and Discussion section, the mechanical property improvements are compelling. The authors have claimed that the test sample was printed as 1BA dumbbell-shaped splines according to the China Standard GB/T 1040.2-2006. Please add CAD model of the test sample.
4. Figure 5 needs more clarity. Instead of qualitative analysis, a quantitative analysis would validate the claim on 4D printable resin.
5. The manuscript discusses the potential for commercial applications but does not address scalability challenges and the feasibility of industrial-scale production.
Reviewer 3 Report
Comments and Suggestions for Authors
The authors submitted a manuscript entitled „4D printable photocurable resin composed of acrylate waste cooking oil and triethylene glycol dimethacrylate: An effective approach to enhance the tensile strength of resins derived from 4 waste cooking oil“.
The submitted manuscript has clear objectives and novelty of experimentally verified results.
The abstract is good and does not need to be improved.
The introduction is well written and the authors have used appropriate modern literature. The introduction could be shorter.
Graphically, the manuscript is of a good standard.
Conclusion and future work are good.
References are relevant to the topic; the manuscript does not contain self-citations.
The manuscript is well written, with interesting and novel findings that have application potential.
Well done research.
I have no further recommendations.
Overall, it deserves readers' attention and I recommend it for publication in Molecules.
Round 2
Reviewer 2 Report
Comments and Suggestions for Authors
Additional comments regarding previous comment 2:
Author response: "Based on your suggestion, we have supplemented the shape memory cycle of a rectangular thin film printed with TEGDMA-EWOA resin, with experimental results shown in Figure 5a. We have also discussed the shape memory of the rectangular thin film."
The reviewer is concerned about the shape recovery effect. The authors have claimed shape recovery from -60 degrees Celsius to room temperature. Is this effect due to freezing/supercooling, i.e., glass transition? Does this material also show shape recovery at positive temperature ranges?
Author Response
Dear Reviewer,
I am writing regarding to our manuscript entitled “A 4D printable photocurable resin derived from waste cooking oil with enhancing tensile strength” (molecules-2959837-R1). We would like to thank for the precious comments from you. Based on your comment and suggestion, the manuscript has been revised. The modified parts are highlighted in yellow. The responses to your comment are listed below.
(1) Additional comments regarding previous comment 2:
Author response: "Based on your suggestion, we have supplemented the shape memory cycle of a rectangular thin film printed with TEGDMA-EWOA resin, with experimental results shown in Figure 5a. We have also discussed the shape memory of the rectangular thin film."
The reviewer is concerned about the shape recovery effect. The authors have claimed shape recovery from -60 degrees Celsius to room temperature. Is this effect due to freezing/supercooling, i.e., glass transition? Does this material also show shape recovery at positive temperature ranges?
Our response: The shape memory behavior of TEGDMA-EWOA resin was closely related to its glass transition. The room temperature (25 ℃) was higher than the glass transition temperature of TEGDMA and EWOA segments, so the entire WCO-based polymer network exhibited good elasticity and flexibility. Its original shape could be easily changed under external forces, thereby forming temporary deformation shapes. When products with temporary deformation shapes were cooled at a lower fixed temperature (such as −60 °C), the TEGDMA and EWOA segments could be well "frozen," and the entire system transformed into a hard and brittle state. From a macro perspective, the temporary deformed shape could not deform again at the fixed temperature, thus achieving good shape fixation. Then, if the product with a fixed deformation shape was placed at a relatively high recovery temperature (such as 25 °C), the TEGDMA and EWOA segments unfroze and regained their elasticity and flexibility, resulting in full shape recovery and completing a shape memory cycle.
To ensure the shape memory effect of the resin, the operating temperature could be determined based on DSC data. the DSC curve of A2 resin exhibited an exothermic peak at −31.9 °C, indicating that below this temperature, the entire polymeric network of the resin was in a "fully frozen" glass state. Therefore, to ensure effective shape fixation, the material should be kept deeply subcooled, namely at a suitable temperature below −31.9°C. For instance, A2 resin displayed a high Rf (98.33%) when fixed in a dry ice-ethanol solution at −60 °C. Conversely, if a higher temperature (> −31.9°C) was chosen, some segments of the A2 resin chains thaw partially, leading to incomplete fixation. For example, its Rf decreased to 90.55% and 79.00% at −25 °C and 0 °C, respectively. On the other hand, during the heating curve, an endothermic peak at 21.65 °C was observed, which could be attributed to the glass transition of TEGDMA and EWOA segments, indicating complete unfrozen of segments in A2 resin above this temperature. Therefore, a recovery temperature of 25 °C was preferred to ensure complete shape recovery. For instance, A2 resin exhibited a rapid recovery rate during U-shaped testing at 25 °C, with an Rr of 87.5% achieved within 20 seconds and 99.88% within 180 seconds. However, when shape recovery was attempted at lower temperatures, some "frozen" WCO and TEGDMA segments were unable to fully move, resulting in lower recovery rates. At −25 °C and 0 °C, the material only achieved Rr values of 30.56% and 58.33%, respectively. Conversely, at higher temperatures (e.g., 50 °C) in hot water, A2 resin could achieve complete recovery at a much faster rate, reaching an Rr of 99.00% within 4 seconds. However, at this temperature, the mechanical performance of A2 resin significantly decreased, thus caution was required during operation to avoid product damage.
Based on your suggestion, we have added the recovery data of A2 resin at 50 °C. Please refer to the revised Figure 6h.
Thank you very much for your help.